# CKG: Improving ABSA with text augmentation using ChatGPT and knowledge-enhanced gated attention graph convolutional networks

**Yapeng Gao, Lin Zhang** *, **Yangshuyi Xu**

College of Information Engineering, Shanghai Maritime University, Shanghai, China

* linzhang@shmtu.edu.cn

## Abstract

Aspect-level sentiment analysis (ABSA) is a pivotal task within the domain of neurorobotics, contributing to the comprehension of fine-grained textual emotions. Despite the extensive research undertaken on ABSA, the limited availability of training data remains a significant obstacle that hinders the performance of previous studies. Moreover, previous works have predominantly focused on concatenating semantic and syntactic features to predict sentiment polarity, which inadvertently severed the intrinsic connection. Several studies have attempted to utilize multi-layer graph convolution for the purpose of extracting syntactic characteristics. However, this approach has encountered the issue of gradient explosion. This paper investigates the possibilities of leveraging ChatGPT for aspect-level text augmentation. Furthermore, we introduce an improved gated attention mechanism specifically designed for graph convolutional networks to mitigates the problem of gradient explosion. By enriching the features of the dependency graph with a sentiment knowledge base, we strengthen the relationship between aspect words and the polarity of the contextual sentiment. It is worth mentioning that we employ cross-fusion to effectively integrate textual semantic and syntactic features. The experimental results substantiate the superiority of our model over the baseline models in terms of performance.

## 1. Introduction

In recent years, with the pervasive utilization of social media and online reviews, individuals have been afforded greater opportunities to articulate their emotions on the internet [1]. These expressions of emotion not only involve overall sentiment polarity determination but also encompass specific aspects of emotions. This presents new challenges for neurorobotics in understanding textual emotions at a more intricate level. Therefore, research on ABSA has become increasingly important. ABSA is a branch of sentiment analysis that aims to identify and analyze the sentiment orientation towards specific aspects or targets in text. As shown in the Fig 1, in product reviews, users may have different sentiment evaluations for different aspects of the product, such as performance, design, price, and so on. Understanding these aspect-level sentiment information can help companies gain insights into consumer satisfaction with different aspects, thereby improving products and providing better user experiences.

2004], REST15 [https://aclanthology.org/S15-2082], REST16 [https://aclanthology.org/S16-1002], and X (Twitter) [https://aclanthology.org/P14-2009].

**Funding:** The work has supported by the Shanghai Sailing Program [21YF1416700]. The funders had no role in study design, data collection and analysis, decision to publish, or preparation of the manuscript.

**Competing interests:** The authors have declared that no competing interests exist.

However, ABSA faces several challenges. Aspect identification is a critical issue because aspects are often specific entities, attributes, or viewpoints that are contextually related [2]. Furthermore, ABSA needs to consider the sentiment expressions associated with different aspects in the text, such as positive, negative, or neutral sentiments. The challenge in this multi-label classification problem lies in accurately capturing the sentiment information related to aspects. Additionally, sentiment expressions in the text may be implicit or ambiguous, requiring context understanding and inference.

Most of the early work on ABSA utilized neural network models to extract sentiment information from the given aspect in context. Although earlier models based on temporal models [3–6] and attention mechanisms [7, 8] have achieved certain effectiveness, they still face several challenges, including the scarcity of sufficient training data, imbalanced data distribution, and limitations in the generalization and practical applicability of existing models. Consequently, the analysis of syntax has gradually gained increasing attention, as shown in the Fig 2. In addition, existing methods still need improvement in handling grammatical and semantic learning abilities, as well as effectively linking aspect words, context, and multiple network features.

The scarcity of training data and the imbalanced distribution of data present significant challenges in ABSA. Obtaining domain-specific data is often a complex task, resulting in limited training data size. This limitation can undermine the generalization ability of the models, hindering their performance across diverse scenarios. Furthermore, the uneven distribution of data introduces instability in model performance when confronted with various aspect and sentiment expressions. Moreover, existing methods exhibit limitations in their grammatical and semantic learning abilities. Traditional attention mechanisms excel at extracting semantic information but struggle with handling grammatical nuances effectively. Conversely, methods solely reliant on grammar dependency trees fail to capture the contextual information of aspect words adequately. As a result, the model's understanding capability within complex contexts becomes restricted. Addressing these limitations is imperative to enhance the grammatical and semantic learning abilities of ABSA models.

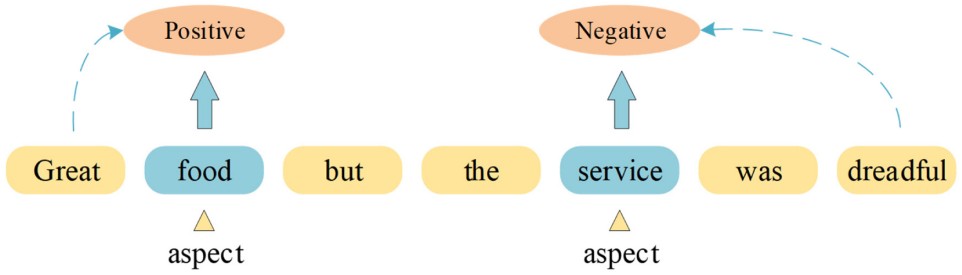

**Fig 1. Example of aspect-level sentiment analysis.**

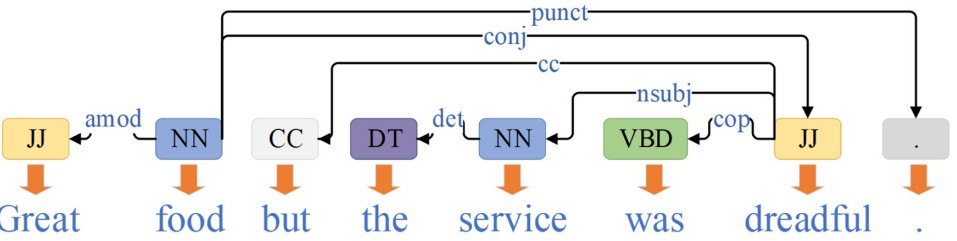

**Fig 2. Dependency tree of a sentence.**

To tackle the aforementioned challenges, this study presents the CKG (Aspect-Level Sentiment Analysis with Text Augmentation using ChatGPT and Knowledge-Enhanced Gated Attention Graph Convolutional Network) model. The CKG model employs a graph convolutional neural network to analyze sentences, leveraging sentiment knowledge from SenticNet to amplify the influence of aspect words on sentiment polarity within the sentence's dependency graph. Furthermore, the model incorporates gate attention mechanisms to update nodes. In comparison to conventional graph neural networks, the CKG model exhibits enhanced efficiency and superior performance. Additionally, we streamline the attention mechanism by employing relative position encoding to extract semantic information, thereby effectively handling the relative relationships of positional information within sequences.

Furthermore, before the final sentiment prediction output, we employ cross-fusion to integrate the final outputs of the sequence module and the graph convolution module. Compared to previous concatenation or summation methods, this approach not only improves the model's performance but also enhances its interpretability. In summary, CKG not only pays attention to the syntactic dependency of texts but also effectively combines the semantic features of neighboring nodes, thereby enhancing the model's understanding of text semantics and grammatical structures.

The main contributions of our work can be summarized as follows:

- A method is proposed to expand text data based on the original dataset using ChatGPT.

- A graph convolutional network with gate attention mechanisms is introduced, which enhances the aspect features of the original sentence by incorporating syntactic dependency trees and an external knowledge base. This ensures the full consideration of the connection between the original sentence and aspect words, avoiding the loss of information.

- We propose a CKG model, which integrates syntactic and semantic features through cross-fusion, and its effectiveness in ABSA tasks has been demonstrated on five benchmark datasets.

## 2. Related works

### 2.1 Aspect-based sentiment analysis

Aspect-based sentiment analysis (ABSA) is an important research direction in the field of sentiment analysis. Its goal is to identify and analyze the sentiment inclination towards specific aspects or targets in the text. This task is more fine-grained, judging the sentiment polarity of the text based on specific aspect words. Compared to document-level or sentence-level sentiment analysis, ABSA is more reasonable and practical [9]. The key to ABSA is to accurately identify and analyze the sentiment inclination towards specific aspects or targets in the text. ABSA needs to focus on aspect words and sentiment words in the text and understand the relationship between them. When conducting sentiment analysis, it is necessary to consider the semantic information of the context and the complexity of the sentiment expression. Traditional machine learning algorithms require manual design and selection of features, such as word bag models and n-gram features. The selection of these features may be influenced by subjective factors and cannot fully capture the semantic information of the text. In addition, these methods require a large amount of annotated data to train the model, and the acquisition cost of annotated data is high. When the dataset changes, the performance of the methods is also greatly affected [10]. Therefore, at the current stage, this technology is basically only used as a research reference.

With the advancement of deep learning technology, models based on Convolutional Neural Networks (CNN) and Recurrent Neural Networks (RNN) have been widely used in this task. These models can learn richer feature representations from the text and capture the complex relationship between sentences and aspects. In addition, the development of word embedding techniques, such as Word2Vec [11], PV [12], and GloVe [13], has also promoted the progress of deep learning in sentiment analysis tasks. Long Short-Term Memory (LSTM), a variant of Recurrent Neural Network (RNN), is widely used in sentiment analysis tasks, especially in solving the problem of long-term dependencies in time sequences. However, traditional LSTM has some limitations, such as being difficult to train in parallel and ineffective in handling the joint relationship between context information and aspect words in ABSA, ignoring their intrinsic connection.

## 2.2 Attention mechanism in aspect-based sentiment analysis

In recent years, attention mechanisms have received considerable attention in ABSA research because they can capture the relationship between aspects and sentiments and place them in a context [14]. Attention mechanisms allow models to focus on relevant parts of the text when generating sentiment predictions, effectively handling long and context-rich sentences. By assigning different weights to each word or phrase, attention mechanisms can emphasize the most informative aspects, improving overall sentiment classification performance.

Some studies have explored the application of attention mechanisms in ABSA and achieved good results. For example, researchers have proposed using self-attention mechanisms to capture the importance of different words in aspect-related contexts [15]. This technique allows the model to assign higher weights to words related to specific aspects, thereby improving ABSA. In addition, hierarchical attention mechanisms [16] have been used to capture sentiment information at different granularity levels. By assigning attention weights at the word level and aspect level, it can effectively capture the sentiment expressed for each aspect and aggregate the information for final sentiment prediction. This approach has shown better performance in capturing subtle differences in fine-grained sentiment.

Despite the notable achievements of attention mechanisms in ABSA, several challenges persist that necessitate attention and resolution. One such challenge pertains to the limited interpretability of attention weights. While attention mechanisms offer insights into the significance of different words or phrases, elucidating the precise reasoning process underlying the assigned weights often proves arduous. Techniques aimed at enhancing the interpretability of attention weights can significantly contribute to understanding the decision-making process of the model. It is noteworthy that the resilience of attention mechanisms against noise and ambiguous inputs remains an open question. Noise inputs encompassing spelling errors, grammatical inaccuracies, or informal language usage can potentially impede the accurate allocation of attention weights and hinder the capture of relevant sentiment information. Research dedicated to developing attention mechanisms resilient to such noise is indispensable for practical applications.

## 2.3 GNN in aspect-based sentiment analysis

To address the limitations of attention mechanisms in ABSA, research based on Graph Neural Networks (GNN) has received widespread attention in the ABSA field. GNN has the ability to model complex relationships and capture contextual information [17], making it a significant application in ABSA. GNN provides a powerful framework for integrating syntactic and semantic information, allowing models to better understand the sentiments expressed for different aspects.

Some studies have explored the application of GNN in ABSA and achieved promising results. Researchers have proposed methods that represent text data as a graph, where aspects and related sentiments are represented as nodes, and their relationships are modeled as edges. By leveraging GNN [18], these models can capture the interactions between aspects and sentiments, thereby improving sentiment analysis performance. Zhang et al. [19] have developed the Tree Communication Model (TCM), which uses graph convolutional networks and recurrent graph neural networks to construct a tree structure based on syntactic parsing results. This model achieves better performance in text sentiment classification by capturing the relationships between nodes in a more comprehensive way. Building on this idea, Zhou et al. [20] have improved the syntax tree structure by learning a tree structure centered on aspects to shorten the distance between aspects and corresponding opinion words, showing good results through experiments. In order to further enhance the impact of aspect words on graph network construction and their relevance to the context, some works [21, 22] have attempted to introduce external knowledge bases to increase the importance of aspect words in constructing graph networks. However, although progress has been made in applying GNN in ABSA, there are still some unresolved research questions, such as how to effectively integrate external knowledge into graph networks, how to effectively apply attention mechanisms to improve the understanding of graph networks, and how to avoid the problem of gradient explosion during training.

In this paper, we propose a Graph Convolutional Network (GAGCN) with gate-controlled attention mechanisms that focus the network's attention on nodes crucial for ABSA, thereby improving the model's performance. Furthermore, we introduce SenticNet [23], a sentiment knowledge base, to enhance the weights of aspect nodes and their related words, ensuring that the model's attention is focused on aspect words and their corresponding sentiment tendencies. Additionally, this method can effectively handle multi-label tasks and achieve sentiment analysis for multiple aspects by assigning different attention weights to different aspects. Experimental results show that the proposed model, which combines sentiment knowledge enhancement and an adaptive attention mechanism, performs better.

## 2.4 Overview of ChatGPT

The rapid development of artificial intelligence has brought many exciting technological breakthroughs, one of which is ChatGPT (Chat-based Generative Pre-trained Transformer) [24]. ChatGPT is a deep learning-based natural language processing model with a wide range of applications and profound impact. It can understand and generate natural language text, making human-computer interaction more natural and fluent. ChatGPT and related models have achieved remarkable results in natural language processing tasks through pre-training and fine-tuning methods. By learning language patterns and semantic information from large-scale corpora, they can generate accurate, coherent, and meaningful answers.

As an innovative natural language processing technology, ChatGPT has a wide range of applications in many fields [25]. First, it plays an important role in virtual assistants and customer services. ChatGPT can interact with users in real-time as an intelligent conversational system, providing personalized information and answering questions. In the field of education, ChatGPT can serve as a personalized learning companion, offering question answering, learning materials, and guidance to improve learning outcomes and engagement [26]. ChatGPT is also applied in areas such as intelligent customer support, automated text generation, information retrieval, and smart homes, providing people with convenient and intelligent services.

The emergence of ChatGPT brings significant value and impact. It provides a new way of human-computer interaction, allowing users to have more freedom in conversation with

computers and enjoy personalized services and customized experiences. Intelligent answering and recommendation systems of ChatGPT can provide users with accurate and high-quality information, improving the efficiency and accuracy of information retrieval. Additionally, ChatGPT provides an important tool for researchers in language learning and natural language processing, promoting the development and innovation in these fields.

Despite the significant progress of ChatGPT, there are also negative evaluations and potential issues [27]. Due to its training on large-scale datasets, ChatGPT may have problems related to information bias and implicit bias. Furthermore, ChatGPT may encounter difficulties in handling complex questions and understanding context, leading to inaccurate answers or lack of logical reasoning. The application of ChatGPT has also raised concerns about ethics and privacy, such as privacy protection of user data and potential risks of misuse. This paper mainly uses ChatGPT for text data augmentation based on the original dataset and does not involve sensitive issues.

## 3. Methods

In this section, we provide a detailed explanation of the proposed graph attention convolutional network, which utilizes ChatGPT for data augmentation and incorporates sentiment knowledge enhancement and gate mechanism. We also provide an introduction to the various details of the overall model.

### 3.1 Overview

As shown in the Fig 3, our proposed model consists of two main components: (1) learning contextual representations using BiLSTM or BERT, which takes the embedding matrix of each sentence as input and outputs the contextual feature representation of the sentence. (2) Feature extraction of the syntactic dependencies of the sentences, where the enhanced representations of the sentences are inputted through the sentiment knowledge base and further feature extraction is performed using the graph attention convolutional network with incorporated gate mechanism. (3) The sentence features obtained from (1) and aspect features are enhanced using the self-attention mechanism with relative positional encoding. The graph information outputted by GAGCN is also fused to extract important sentiment dependencies related to specific aspects. Previous methods that solely relied on sequence networks or attention-based methods to analyze global semantics and aspect features failed to effectively link aspect words with underlying syntactic features of the original text. In addition, most models that employed graph convolutional neural networks only analyzed the syntactic dependencies and overlooked the aspect-sentiment connections contained in the semantic information of the text. Our proposed approach addresses these two shortcomings and strengthens the influence of aspects on the sentence sentiment, thereby improving the accuracy and interpretability of sentiment polarity output.

### 3.2 Definition

Aspect-level sentiment classification tasks typically involve preparing an input sequence for the model consisting of a context sequence and an aspect sequence. This input format enables the model to learn the relationship between the context and the aspect term. Assuming that the context sequence containing the input aspect term is denoted by $s = \{w_0, w_1, \ldots, w_n\}$, it contains $n + 1$ words, including the target aspect term. The target aspect sequence is represented as $s^t = \{w_0^t, w_1^t, \ldots, w_m^t\}$ and is a subsequence of $s$ composed of $m(m \geq 1)$ words. The objective of aspect-based sentiment analysis is to determine the sentiment polarity related to the aspect based on the given sentence and aspect term.

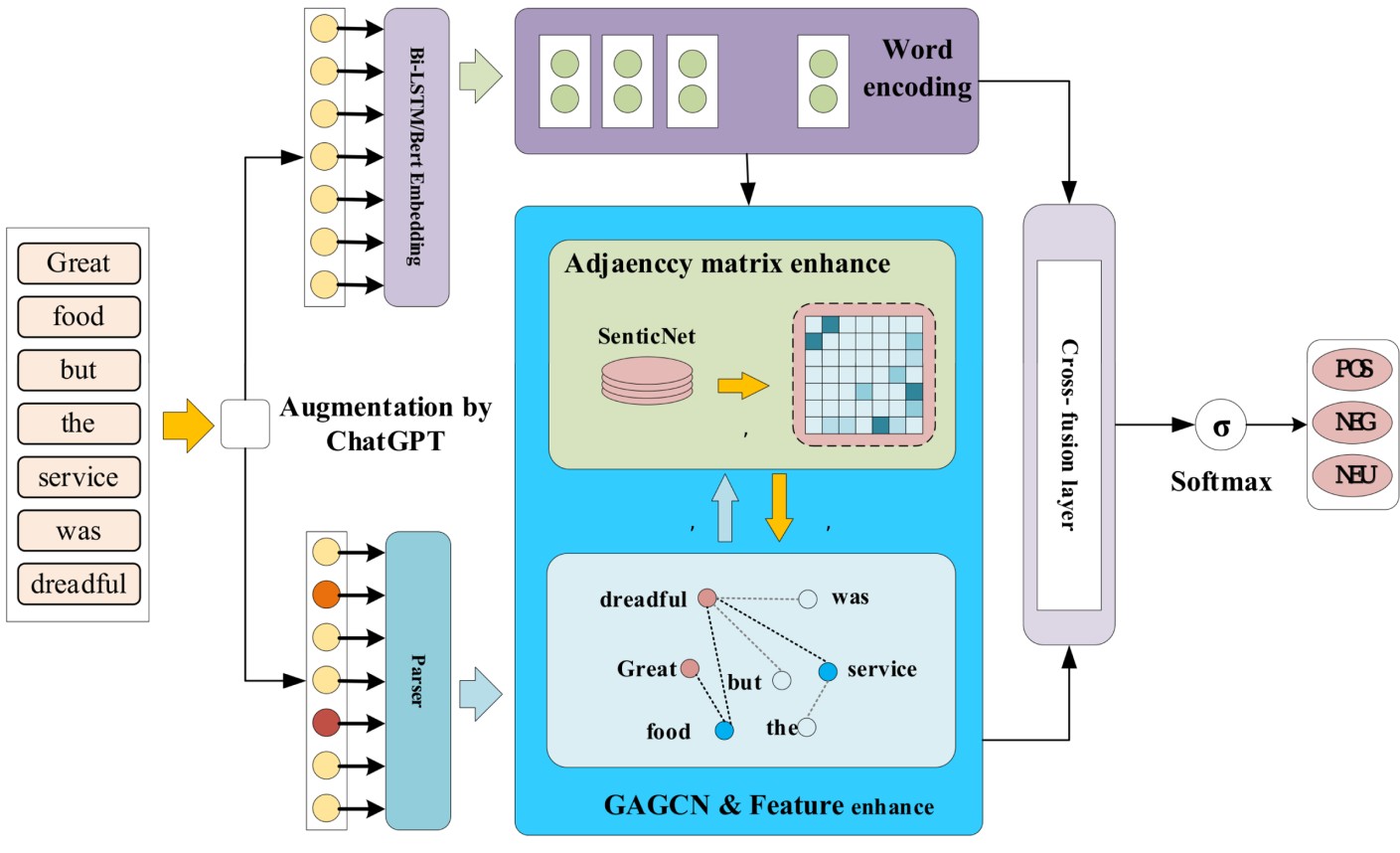

**Fig 3. Overall architecture diagram of the CKG model.**

### 3.3 Data augmentation using ChatGPT

ChatGPT is developed based on the research of GPT [28], GPT-2 [29], and GPT-3 [30]. Its core idea is to incorporate a reinforcement learning model into GPT-3 to fine-tune the model's output to be more reasonable, accurate, truthful, and harmless [31]. During the pre-training phase of ChatGPT, an unsupervised task is performed where input samples are denoted as $X = \{x_1, x_2, \ldots, x_n\}$, $x_i = (s_1, s_2, \ldots, s_m)$. The Transformer model is trained to obtain corresponding tokens and their associated positional encodings. Representing the trainable parameters as $\theta$, the ultimate goal of the pre-training phase is to maximize the following Eq 1:

$$L(x_i) = \sum_{i=1}^{m} \log P(s_i | s_1, \ldots, s_{i-1}; \theta) \tag{1}$$

After pre-training, a reinforcement learning model with human feedback is used to fine-tune the pre-trained model. Human experts are involved in constructing and evaluating samples with questions and their corresponding answers. These prompted samples are utilized for further training of the model. In addition to fine-tuning, ChatGPT incorporates an additional training of a reward model. This reward model combines the question-answer pairs set by human experts with the model's predicted results and assigns reward scores accordingly, as

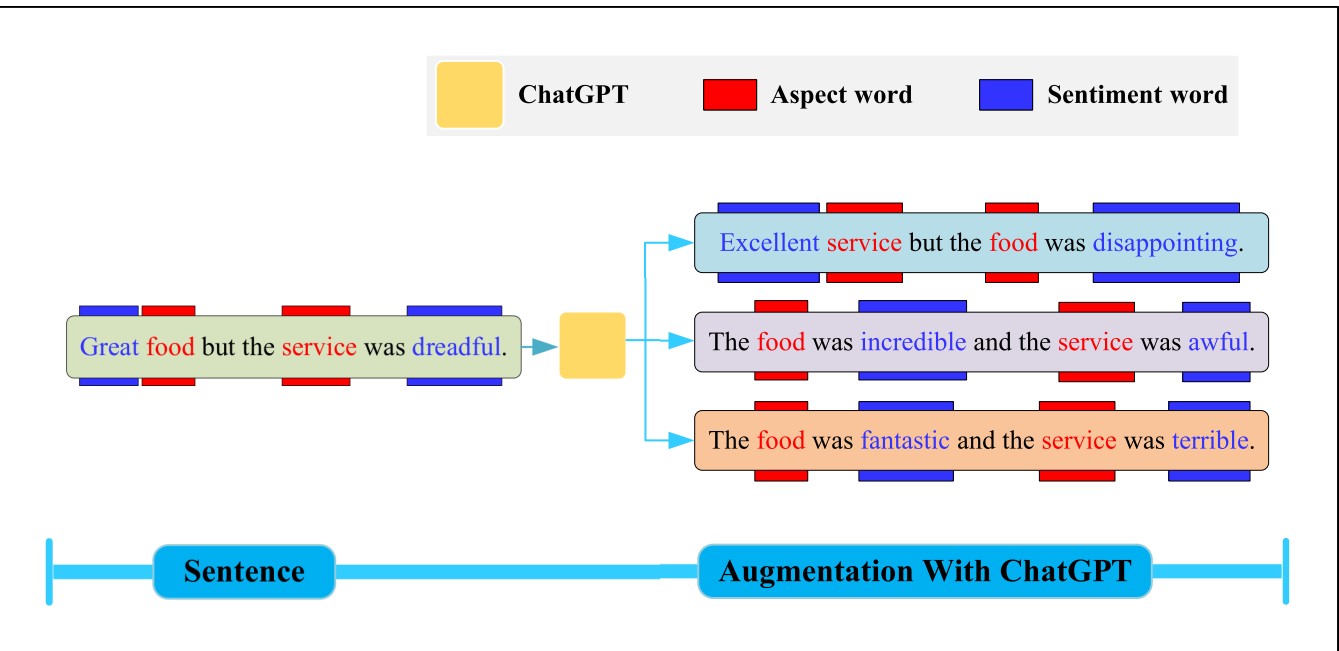

**Fig 4. ChatGPT-generated text examples incorporating aspect words based on the original text.**

shown in the following Eq 2:

$$loss(\theta_r) = E_{(x,y_w,y_l)\sim D_c}[\log(\sigma(r_{\theta_r}(x,y_w) - r_{\theta_r}(x,y_l)))] \qquad (2)$$

Here, $\theta_r$ represents the parameters of the reward model. $x$ denotes the prompt, while $y_w$ represents the preferred completion among the given alternatives in $y_l$. $D_c$ denotes the dataset used for human expert comparisons. Based on the reward model, ChatGPT is fine-tuned using the Proximal Policy Optimization (PPO) strategy, ultimately completing the training of the final model.

There are several models available for text data augmentation [32–34]. However, ChatGPT is particularly suitable for the data augmentation task in this study [35]. This is because ChatGPT combines manually labeled training samples by human experts, resulting in generated content that is more genuine and aligned with human language conventions. Additionally, under the supervision of the reinforcement learning model, the generated content is more reliable and of higher quality. Furthermore, the training of ChatGPT based on a large-scale corpus enhances the diversity of data generation, which is beneficial for expanding the data in this study. Fig 4 presents a real case of text generation by ChatGPT, demonstrating its high-quality output that ensures overall semantic fluency and naturalness of sentences, while also satisfying the diversity requirements of the data augmentation task. Based on ChatGPT, the algorithmic procedure for our data augmentation is outlined as this Algorithm 1.

**Algorithm 1** The framework of ChatGPT for augmenting text in the original dataset.

```
Input: base dataset D and novel Dataset Dₙ
Output: Dₙ
Definition: POS, NEU, NEG ∈ D represent the collections of texts in
the original dataset labeled as positive, neutral, and negative,
respectively, containing aspect words. α ∈ POS, β ∈ NEU, γ ∈ NEG rep-
resent individual texts within these collections. count() denotes a
```

```
counting function, and M is a manually set parameter. ChatGPT_aug(x, y)
represents the data augmentation process using ChatGPT, which takes
two inputs:x ≥ 1, x ∈ N⁺ denotes the number of texts to be generated,
and y ∈ D represents an individual text.
```

1: $M = 3000$

2: **for** $\alpha$ in $POS$ and $count(POS) < M$ **do**

3: $D_{aug}^{POS} = ChatGPT\_aug\left(\lceil \frac{M}{count(pos)}\rceil, \alpha\right)$

4: **end for**

5: **for** $\beta$ in $NEU$ and $count(NEU) < M$ **do**

6: $D_{aug}^{NEU} = ChatGPT\_aug\left(\lceil \frac{M}{count(NEU)}\rceil, \beta\right)$

7: **end for**

8: **for** $\gamma$ in $NEG$ and $count(NEG) < M$ **do**

9: $D_{aug}^{NEG} = ChatGPT\_aug\left(\lceil \frac{M}{count(NEG)}\rceil, \gamma\right)$

10: **end for**

11: $D_n = D_{aug}^{POS} \cup D_{aug}^{NEU} \cup D_{aug}^{NEG} \cup D$

12: **return** $D_n$

The function *ChatGPT_aug*() in Algorithm 1 represents the data augmentation of training text data using ChatGPT. It is worth noting that the prompt configuration has a significant impact on the generation effectiveness of ChatGPT. In Fig 5, we illustrate how our carefully designed prompt effectively guides ChatGPT in performing the data augmentation task.

This paper [35] have employed cosine similarity and transRate to evaluate the quality of generated text. However, we argue that these approaches are not suitable for ABSA. Firstly, when proper prompts are provided and the generation process is carried out iteratively with a limited amount of text, ChatGPT exhibits commendable text generation quality. With the support of large language models, ChatGPT can creatively generate text based on the original input. It may produce text that meets the requirements of ABSA, but differs significantly in similarity to the original sentence. For example, it may change aspect words or alter sentence expressions, resulting in a low cosine similarity Table 3. Therefore, we adopt human expert to assess the quality of the generated text, as it provides a more reliable and accurate measure. To ensure the effectiveness of the augmented data and the feasibility for model training, we randomly selected 50 instances of generated data. These instances were then mixed with 50 instances of real data from the dataset. The combined set was distributed to five human experts specializing in ABSA for scoring tests. Each expert was required to assign a score ranging from -1 to 3 (integer values only) to indicate the suitability of each sentence for ABSA. The specific scoring rules were as Table 1.

## 3.4 Embedding module

The Embedding layer serves as the fundamental building block of the model architecture, and in this study, two methods are employed for text encoding. The first method leverages a Bi-LSTM to compute the textual representation by utilizing word vectors derived from the input text. Conversely, the second method incorporates the utilization of the pre-trained BERT model for text encoding. BERT utilizes a transformer structure, which encompasses a self-attention mechanism facilitating bidirectional information flow. By employing a Masked Language Model (MLM) during pre-training, BERT effectively considers both preceding and succeeding context, thereby comprehensively capturing semantic contexts and enhancing the accuracy of information extraction.

We map each word to an m-dimensional word embedding through an embedding table $X \in \mathcal{R}^{m \times |N|}$, resulting in an embedding matrix $X = [x_1, x_2, \ldots, x_{a_1}, x_{a_2}, \ldots, x_{a_k}, \ldots, x_n]$. Here, $x_{a_i} \in \mathcal{R}^m$ represents the word embedding of the aspect term $w_{a_i}$. The dimensionality of the

---

### The Working Principle of Data Augmentation via chatGPT

**Prompt**

The next task will focus on text data augmentation for aspect-level sentiment analysis. Based on the given text "%T%", %N% data samples will be generated, returned in JSON format, and stored in %Path%. Here is an example of the generated data:{"text": "The location of this hotel is excellent, but the facilities are a bit outdated.","aspect": "location","sentiment": "1"} {"text": "The location of this hotel is excellent, but the facilities are a bit outdated.","aspect": "facilities","sentiment": "-1"
}
(Note: %T%, %N%, and %Path% are placeholders that should be replaced with actual values in the translation.)

**ChatGPT answer**

```
{
  "text": "The screen of this phone is clear and impressive, and the performance is outstanding, but the battery life is disappointing.",
  "aspect": "battery life",
  "sentiment": "-1"
}
{
  "text": "The screen of this phone is clear, and the performance is disappointing, but the battery life is excellent.",
  "aspect": "performance",
  "sentiment": "-1"
}
```

**Fig 5. Process flow diagram for text augmentation using prompts in ChatGPT.**

**Table 1. The scoring criteria for evaluating the applicability of text data generated by ChatGPT.**

| Score | Rules |
|---|---|
| -1 | logic inconsistent, semantics unclear, no aspect words |
| 0 | logic inconsistent, semantics ambiguous, no aspect words |
| 1 | logic indistinct, semantics ambiguous, no aspect words |
| 2 | logic distinct, semantics ambiguous, contain aspect words |
| 3 | logic is consistent, semantics clear, contain aspect words |

word vectors is denoted by $m$, $n$ represents the sentence length, and $|N|$ denotes the vocabulary size. We derive the embedding lookup table from pre-trained embeddings such as GloVe or BERT [36], and fine-tune it during the training process.

CKG conducts experimental comparisons between the two approaches. The LSTM-based model demonstrates lower accuracy compared to BERT, but requires significantly less training time. On the other hand, the BERT-based model exhibits significantly improved accuracy at the cost of longer training time and higher computational expenses.

## 3.5 Knowledge-enhanced GCN with attention and gated skip-connection (GAGCN)

The graph structure incorporates syntactic encoding as input. To encode syntactic informa-tion, the SpaCy (The spaCy toolkit is used to parse the dependency tree of the sentence: https://spacy.io/.) dependency parser is used to compute syntactic dependencies. By providing the syntactic structure, we capture richer information about the sentence structure and con-struct the adjacency matrix $D \in \mathbb{R}^{n \times n}$ according to Eq 3:

$$D_{i,j} = \begin{cases} 1 & if \ w_i, w_j \ contains \ dependency \\ 0 & otherwise \end{cases} \tag{3}$$

To leverage the relationships between words, our graph structure adopts an undirected dependency graph. Inspired by [37], we enhance the adjacency matrix with sentiment infor-mation using SenticNet. First, we need to calculate the SenticNet scores between nodes, which can be represented by the following Eq 4:

$$S_{i,j} = SenticNet(w_i, w_j) \tag{4}$$

Where, $SenticNet(w_i, w_j) \in [-2, 2]$ represents the combined sentiment score of $w_i$ and $w_j$ after being computed by SenticNet. This Eq 4 ensures that the model tends to favor words with higher sentiment scores, capturing more accurate sentiment polarity. It is worth noting that if $w_i$ does not appear in SenticNet or is a neutral word, $SentNet(w_i) = 0$. Next, based on the com-puted $S_(i, j)$, we can enhance the previously constructed adjacency matrix with sentiment information using the following Eq 5:

$$A_{i,j} = D_{i,j} \times (S_{i,j} + T_{i,j} + 1) \tag{5}$$

$$T_{i,j} = \begin{cases} 1 & if \ w_i \ or \ w_j \ is \ an \ aspectword \\ 0 & otherwise \end{cases} \tag{6}$$

Using the adjacency matrix $A_{i,j}$ for syntactic encoding, GAGCN takes the input hidden state vectors $H$ as initial node representations in the syntactic graph. Then, the GAGCN module obtains the graph representation of the syntax as $H^{syn} = \{h_1^{syn}, h_2^{syn}, \ldots, h_n^{syn}\}$, where the symbol $h_i^{syn} \in \mathcal{R}^{d_u}$ represents the hidden representation of the $i$-th node. The update equation for the representation of the $i$-th node in the $l$-th layer is as Eq 7:

$$h_i^{synl} = \phi(\sum_{j=1}^{n} A_{i,j} \mathcal{F}(h_j^{synl-1} + n_{att}^i) W^l + b^l) \tag{7}$$

$W^l$ represents the weight matrix and symbol $b^l$ represents the bias term. For aspect nodes, we use the $\{h_{a_1}^{syn}, h_{a_2}^{syn}, \ldots, h_{a_n}^{syn}\}$ to represent their hidden representations. $F$ is the position-

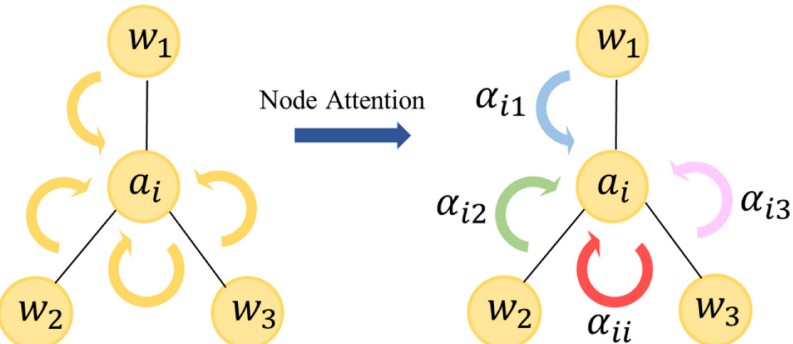

**Fig 6. The arrows represent the information propagation between nodes, and the different colors of the arrows indicate the varying importance of adjacent nodes to the aspect term node $a_i$.**

aware function referenced from [38], and $\theta$ denotes the relu activation function. Based on this equation, assuming there exists a node $H_i^l$ in the $l$-th layer with three neighboring nodes $H_1^l$, $H_2^l$, and $H_3^l$, we can simplify and rewrite the above equation by omitting the bias term as Eq 8:

$$sH_i^{l+1} = \phi(H_i^l W^l + H_1^l W^l + H_2^l W^l + H_3^l W^l) \tag{8}$$

Based on this equation, we introduce the attention mechanism as shown in the Fig 6 to update the node states as Eq 9:

$$H_i^{l+1} = \phi(\alpha_{ii}^l H_i^l W^l + \alpha_{i1}^l H_1^l W^l + \alpha_{i2}^l H_2^l W^l + \alpha_{i3}^l H_3^l W^l) \tag{9}$$

Where $\alpha_{ij}^l$ represents the attention score for the $j$-th node influencing the update of the $i$-th node in the $l$-th layer. The general attention calculation can be simply represented by the Eq 10:

$$\alpha_{ij}^l = \phi(H_i^l W^l, H_j^l W^l) \tag{10}$$

From the above equation, we can deduce the attention weights between each node and its neighboring nodes. However, for ABSA, inspired by [39], our focus should be on the influence of aspect terms on the sentence rather than the attention weights between adjacent nodes. Therefore, we can evaluate the attention coefficients by calculating the coupling between aspect terms and other words.

$$\alpha_{ij}^l = tanh((H_i^l W^l)C^l, (H_j^l W^l)^T) \tag{11}$$

It is worth noting that Zhang in [38] employed Aspect-aware Attention to retrieve important features semantically related to the aspect terms from the hidden state vectors. Accordingly, attention weights based on retrieval were assigned to each context word. Inspired by this, we propose the aspect term weight anchoring method. This method utilizes aspect term-specific masking to mask out non-aspect words learned by the final GCN layer's output, while keeping the representation of aspect terms unchanged. This is done to construct our coupling matrix. Since we enhance aspect terms in the graph network with SenticNet, it is necessary to anchor the representation of aspect terms to prevent the model from weakening the sentiment representation of aspect terms during the training process. This is shown in the following

equation:

$$\dot{h}_t = \begin{cases} \dot{h}_t & if \eta \leq t \leq \eta + l_a \\ 0 & otherwise \end{cases} \tag{12}$$

Where $\dot{h}_i$ represents the representation of the $i$-th word obtained through GAGCN, $t$ is the starting index of the aspect term in the sentence, and $l_a$ is the length of the aspect term. From this, we can obtain the final representation of the anchored aspect term masking:

$$\dot{C}_{mask}^{synl} = \{0, \ldots, \dot{h}_\eta, \ldots, \dot{h}_{\eta+l_a-1}, \ldots, 0\} \tag{13}$$

Furthermore, we utilize a multi-head attention mechanism to describe the interactions between nodes using $K$ different channels:

$$H_i^l = tanh\left(\frac{1}{K}\sum_{k=1}^{K}\sum_{j\in N(i)}\alpha_{ij,k}^l H_j^l W^l\right) \tag{14}$$

As is well known, when multiple graph convolutional layers are added, the model can better understand long-distance information. However, issues such as gradient explosion may arise, which can impact the performance of the model. Therefore, we introduce a gate mechanism in skip connections to accurately propagate historical information when updating the hidden states, as shown in the Fig 7. This gate mechanism determines the forget rate and update rate, ensuring precise information transfer.

$$H_{i,gsc}^{l+1} = \gamma_i \odot H_i^{l+1} + (1 - \gamma_i) \odot H_i^l \tag{15}$$

$$\gamma_i = \sigma(G_{\gamma,1}H_i^{l+1} + G_{\gamma,2}H_i^l + b_\gamma) \tag{16}$$

Where $G_{\gamma,1}$, $G_{\gamma,2}$ and $b_\gamma$ are trainable parameters, $\odot$ represents element-wise matrix multiplication, and $\sigma$ denotes the sigmoid activation function.

Finally, in order to effectively connect the semantic module and the syntactic module features and fully consider all information, we employ cross fusion [40] as the feature fusion

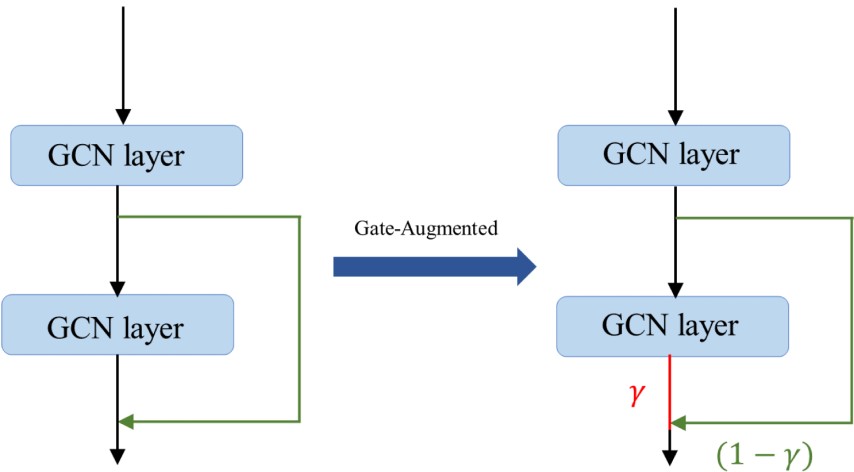

**Fig 7. The difference between skip-connection and gated skip-connection.**

method, calculated through the following equation:

$$H^{syn'} = f(H^{syn} W_1 (X)^T) X \tag{17}$$

$$H^{sem'} = f(X W_2 (H^{syn})^T) H^{syn} \tag{18}$$

Where $W_1$ and $W_2$ are trainable parameters, and $f$ represents the softmax activation function. After this operation, both $H^{syn'}$ and $H^{sem'}$ fully consider the features of syntax and semantics. Then, the final fusion feature $r$ is calculated by applying average pooling (ap) to these features:

$$h_a^{syn} = ap(h_{a_1}^{syn} h_{a_2}^{syn}, \ldots, h_{a_m}^{syn}) \tag{19}$$

$$h_a^{sem} = ap(h_{a_1}^{sem} h_{a_2}^{sem}, \ldots, h_{a_m}^{sem}) \tag{20}$$

$$r = [h_a^{syn}; h_a^{sem}] \tag{21}$$

## 3.6 Output layer

The CKG will extract features from the global structural information, semantic information, and the content of the target aspect. The final representation $r$ is then sent to a linear layer and passed through the softmax function to generate the probability distribution $y$ for the given aspect $a$:

$$y = softmax(W_p r + b_p) \tag{22}$$

Here, $W_p$ and $b_p$ represent the learnable weights and biases, respectively. The function *softmax* () denotes the softmax function, which enables us to learn the final distribution of emotions as output.

## 3.7 Model training

We employ the standard gradient descent algorithm to optimize and update the parameters of the proposed model. The objective of training the model is to minimize the cross-entropy loss through $L_2$ regularization:

$$loss = -\sum_{i=1}^{s}\sum_{j=1}^{c} \hat{y} log y_i^j + \lambda \|\Theta\|^2 \tag{23}$$

Here, $S$ represents the number of training samples, and $C$ denotes the number of classes. $\hat{y}$ represents the correct distribution of emotions. $\Theta$ signifies all trainable parameters. $\lambda$ is the coefficient for the $L_2$ regularization term.

## 4. Experiments

In this section, we will introduce the dataset used to evaluate the model's performance, provide an overview of the experimental parameters, and conclude with a comparative analysis of the models.

## 4.1 Datasets and augmentation with ChatGPT

We trained and validated our model on five publicly available benchmark models: Twitter, proposed by Dong [41], which contains Twitter post data, and four others (LAP14, REST14,

**Table 2. The data distribution of the dataset.** Here, "⇒" represents data augmentation performed using ChatGPT.

| Dataset | Division | Positive | Neural | Negative |
|---|---|---|---|---|
| Twitter | Train | 1507⇒3014 | 3016 | 1528⇒3056 |
| | Test | 597 | 190 | 38 |
| Lap14 | Train | 994⇒3976 | 464⇒3248 | 870⇒3480 |
| | Test | 341 | 169 | 128 |
| REST14 | Train | 2164⇒4328 | 637⇒3185 | 807⇒3228 |
| | Test | 728 | 169 | 128 |
| REST15 | Train | 1178⇒3534 | 50⇒3000 | 382⇒3056 |
| | Test | 439 | 35 | 328 |
| REST16 | Train | 1620⇒3240 | 88⇒3080 | 709⇒3545 |
| | Test | 597 | 38 | 190 |

REST15, REST16) from the lap and restaurants domains of SemEval 2014 task 4 [42], the restaurants domain of SemEval 2015 task 12 [43], and the restaurants domain of SemEval 2016 task 5 [44]. Based on these five datasets, we performed data augmentation operations of varying scales. The aim was to balance the distribution of different sentiment polarities and avoid data inconsistencies, thereby improving the training effectiveness and robustness of the model. Through the algorithm 1, we obtained the augmented datasets, as shown in Table 2. Table 3 presents examples of the augmented data. Upon observation, we found that the quality of the generated data is high, to the extent that aspect terms can be replaced without contradicting the final sentiment polarity.

**Table 3. Examples of the original text after data augmentation using ChatGPT is as follows.** We implement the calculation of cosine similarity using the sklearn package [45].

| Original Sentence | Text augmented via ChatGPT | Cosine Similarity |
|---|---|---|
| The plot of this movie is excellent. | The cinematography of this movie is excellent, capturing the plot perfectly. | 0.758 |
| | The character development in this movie is excellent, enhancing the overall plot. | 0.571 |
| | The pacing of this movie is excellent, keeping the plot engaging from start to finish. | 0.619 |
| The service at this restaurant is terrible, but the food is delicious. | Despite the terrible service, the ambiance and decor of this restaurant create a pleasant dining experience. | 0.323 |
| | While the service may be lacking, the presentation of the food is exceptional and adds to the overall dining experience. | 0.384 |
| | Although the service leaves much to be desired, the extensive menu and the quality of the food make up for it. | 0.286 |
| The screen of this phone is clear, but the performance and battery life are disappointing. | Despite the disappointing performance and battery life, the clear screen of this phone provides an enjoyable visual experience. | 0.617 |
| | The screen clarity of this phone is commendable, but the disappointing performance and battery life hinder the overall user experience. | 0.692 |
| | While the screen is clear and vibrant, the disappointing performance and short battery life limit the phone's functionality. | 0.630 |

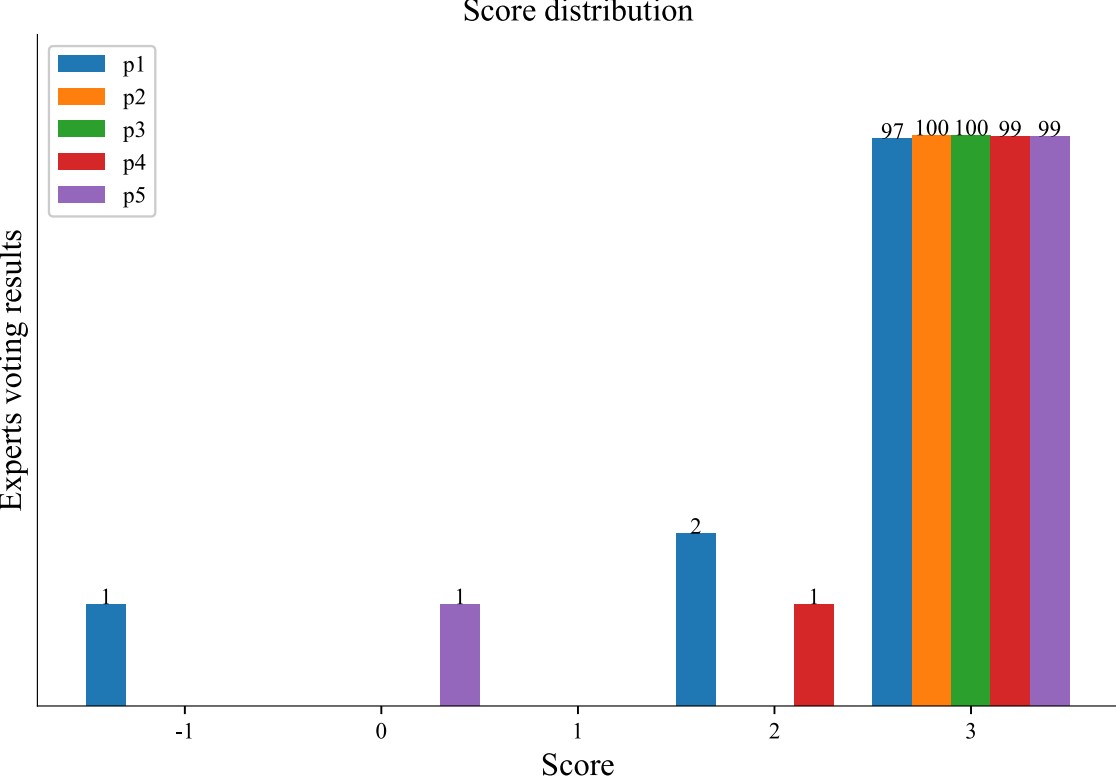

**Fig 8. The statistical representation of the scoring results by human experts.** The experts (p1 to p5) are represented on the x-axis, while the y-axis represents the number of votes corresponding to each score level. Due to the significant disparity in the data distribution, we have applied the operation $\log(y) + 1$ to the y-axis during plotting. However, the values indicated on the graph still represent the original y-values.

In order to evaluate the suitability of the generated samples for ABSA tasks, we conducted an evaluation of the generated data based on the assessment rules outlined in section 3.3, which is shown in Table 1. Both the original samples and the generated samples were randomly selected and mixed in a shuffled manner. Five human experts were tasked with scoring each sample based on the specified criteria. The scoring results are presented in the Fig 8.

From Fig 8, it is evident that the majority of human experts considered these data to meet the requirements of ABSA. These 100 samples were a combination of real and generated data, further demonstrating the effectiveness of data augmentation using ChatGPT for training models in ABSA. Interestingly, certain experts assigned scores of -1, 0, and 2 to specific samples. We extracted these samples and compiled them in Table 4.

**Table 4. Several examples with lower scores.**

| Sentence | Score |
|---|---|
| The bookshelf sprouted wings and flew through the library, while the fish swam through the pages of a novel. | -1 |
| The pencil drew swirling galaxies, while the teacup recited mathematical equations in a hushed tone. | 0 |
| The plot of this novel is so complex, it's as simple and understandable as a clear glass of water. | 2 |
| The quality of this product is truly indescribable, it's like a leaping deer, full of stability and reliability. | 2 |
| The coffee mug floated through the sky, carrying dreams and memories on the wings of a butterfly. | 2 |

Upon analyzing the original sentences, it becomes apparent that these sentences express abstract semantics or logic, making them less suitable for ABSA. However, they possess an artistic quality in terms of their sentence structure. After analysis and adjustment, we concluded that this discrepancy can be attributed to the "memory" function of ChatGPT, which requires further improvement. Generating the entire batch of texts at once can cause ChatGPT to "forget" the initial requirements. Therefore, it is advisable to generate the data in batches. Through our experiments, we found that controlling the number of generated texts to be within 20 entries or less yields better results.

## 4.2 Models for comparison

To evaluate the performance of our proposed CKG model, we compared it with other existing models:

- SVM [46], a classical model used for addressing ABSA.

- TD-LSTM [3], a model that utilizes an improved version of recurrent neural networks to extract text features.

- MemNet [47], a model that explicitly considers contextual information and utilizes a memory network architecture.

- ATAE-LSTM [48], a model that introduces an aspect-sentence attention mechanism to incorporate aspects and sentences into LSTM, thereby effectively considering the crucial information in the text.

- IAN [4], a method that optimizes the attention mechanism by proposing an interactive attention mechanism, which takes into account the feature representation of aspect words and context, ensuring comprehensive consideration of both aspects in the text.

- GCAE [49], a model based on convolutional neural networks and gating mechanisms, selectively outputs sentiment features based on given aspects or entities. The model structure is lightweight and straightforward.

- ASGCN-DT [38], a model that constructs a directed graph based on dependency trees obtained from sentence analysis. It combines graph convolutional networks to extract syntactic features, thereby integrating semantic and dependency relationship analysis for sentiment information.

- ASGCN-DG [38], a model that shares the same core approach as ASGCN-DT but differs in the construction of the graph. Instead of a directed graph, ASGCN-DG utilizes an undirected graph.

- AOA [50], a model that introduces an attention-over-attention mechanism to learn the interaction between aspect words and sentences, focusing on the important parts of the sentence that have a significant impact on the overall sentiment.

- TransCap [9], a model that introduces a transfer capsule network. It leverages aspect routing methods to encapsulate sentence-level semantic representations from aspect-level and document-level data into semantic capsules, thus enhancing the influence of aspect sentiment on semantics.

- BERT [36], an application of the original BERT model in ABSA, utilizing positional encoding as the input for text.

- AEN-BERT [51], a model that introduces label smoothing regularization and utilizes attention-based encoders.

- MWGCN-BERT [52], a model that addresses the issue of long-distance dependencies by combining the LCG method to create a locally contextualized weighted adjacency graph.

- EK-GCN [53], a model that introduces external knowledge to compensate for the inability of syntactic dependency trees to capture fine-grained labels. It also designs a word-sentence interaction network that fully considers aspect information.

- Dual-GCN [54], an improved model that enhances the dependency parser and jointly considers both syntactic structure and semantic relevance.

- T-GCN [55], a method that explicitly leverages dependency types in ABSA using a type-aware graph convolutional network. It utilizes attention to differentiate between different edges in the graph.

- Sentic-GCN [23], a model that incorporates SenticNet to enhance the dependency graph, considering the dependency relationship between contextual words and aspect words, as well as the sentiment information between opinion words and aspect words.

- KGAN [21], proposes a knowledge graph attention network that utilizes a hierarchical fusion module to capture sentiment feature representations from multiple perspectives.

### 4.3 Evaluation metrics

The evaluation metrics for the model include Accuracy (Acc) and macro-averaged F1-score. Accuracy is a commonly used evaluation measure, which represents the ratio of correctly predicted samples to the total number of samples. Macro-averaged F1-score (F1) is a performance metric for multi-class classification problems. It assesses the performance of the model for each individual class and calculates the average as an overall performance indicator for the model:

$$Acc = \frac{P_{true}}{Total} \tag{24}$$

$$F1 = \frac{1}{N} \sum_{i=1}^{N} \left( \frac{2 \times P_i \times R_i}{P_i + R_i} \right) \tag{25}$$

### 4.4 Overall performance comparison

We present the comparison results of our proposed CKG model with some existing excellent ABSA models in Table 5. The data used in the table is sourced from the reported results in the original papers. Any missing data is indicated with a "-". Based on the displayed results in the table, our model demonstrates excellent performance across all five datasets.

Specifically, the traditional analytical methods exhibit relatively poorer performance. LSTM encoding methods improved with attention mechanisms show some enhancements, but they still fall short compared to models that incorporate GCN due to the absence of syntactic dependency knowledge. When we focus on the models that include GCN, we observe significant improvements in performance, particularly evident in the REST15 and REST16 datasets.

**Table 5. Main experimental results of five datasets.** "Acc" represents accuracy, "F1" represents Macro-F1 score. The best results are shown in bold and second best underlined. The experimental results of other models are partly from the original paper and partly verified through reproducing the open-source code.

| Model | TWITTER | | LAP14 | | REST14 | | REST15 | | REST16 | |
|---|---|---|---|---|---|---|---|---|---|---|
| | Acc | F1 | Acc | F1 | Acc | F1 | Acc | F1 | Acc | F1 |
| SVM | 63.40 | 63.30 | 70.49 | - | 80.16 | - | - | - | - | - |
| TD-LSTM | 70.62 | 69.01 | 71.48 | 68.43 | 78.11 | 66.73 | 78.80 | 61.71 | 83.77 | 71.20 |
| MemNet | 68.52 | 66.71 | 72.34 | 64.25 | 79.13 | 66.41 | 80.12 | 67.46 | 82.58 | 67.32 |
| ATAE-LSTM | - | - | 68.70 | 63.93 | 77.20 | 67.02 | 78.48 | 60.53 | 83.77 | 61.71 |
| IAN | - | - | 72.05 | 67.38 | 79.26 | 70.09 | 78.54 | 52.65 | 84.74 | 55.21 |
| GCAE | - | - | 69.14 | 68.71 | 77.28 | 62.45 | 77.56 | 56.03 | 83.70 | 62.69 |
| ASGCN-DT | 71.53 | 69.68 | 74.14 | 69.24 | 80.86 | 72.19 | 79.34 | 60.78 | 88.69 | 66.64 |
| ASGCN-DG | 72.15 | 70.40 | 75.55 | 71.05 | 71.05 | 80.77 | 79.89 | 61.89 | 88.99 | 67.48 |
| AOA | - | - | 72.62 | 67.52 | 79.97 | 70.42 | 78.17 | 57.02 | 87.50 | 66.21 |
| TransCap | - | - | 73.87 | 70.10 | 79.29 | 70.85 | - | - | - | - |
| BERT | - | - | 77.59 | 73.28 | 84.11 | 76.68 | 83.48 | 66.18 | 90.10 | 74.16 |
| AEN-BERT | - | - | 79.93 | 76.31 | 83.12 | 73.76 | - | - | - | - |
| MWGCN-BERT | 75.00 | 74.30 | 79.78 | 76.68 | 86.612 | 80.18 | 85.61 | 73.32 | 82.05 | 79.21 |
| EK-GCN | 75.84 | 74.57 | 78.46 | 76.54 | 83.96 | 74.93 | - | - | 89.36 | 69.32 |
| Dual-GCN | 76.37 | 75.44 | 82.59 | 79.34 | 88.47 | 82.92 | 81.73 | 65.05 | 89.29 | 68.08 |
| T-GCN | 78.03 | 77.31 | 81.97 | 78.71 | 87.41 | 82.23 | 86.00 | 72.81 | **92.97** | 80.07 |
| Sentic-GCN | - | - | 82.12 | 79.05 | 87.20 | 82.50 | 82.72 | 65.86 | 90.52 | 74.53 |
| KGAN | **79.97** | **79.39** | 82.66 | 78.98 | 87.15 | 82.05 | 86.21 | 74.20 | 92.34 | 81.31 |
| CKG | 77.95 | 77.20 | **83.41** | **79.41** | **89.24** | **83.31** | **88.92** | **76.42** | 92.87 | **81.95** |

Compared to non-BERT-based baseline models, our model performs exceptionally well, reaffirming the effectiveness of our approach. Introducing the BERT-base semantic encoding module leads to significant improvements in our model compared to other models. Experimental results show that our model performs competitively, except for a relatively lower performance on the TWITTER dataset, outperforming the other four datasets. Overall, in ABSA tasks, the CKG+BERT model demonstrates outstanding performance.

## 4.5 Ablation study

To validate the effectiveness of each component in our proposed CKG model, we conducted ablation experiments and present the results in Table 6. We can observe that the models

**Table 6. Experimental results of ablation study on four datasets.** "a" represents the addition of attention mechanism in the regular graph convolution, while "ga" represents the incorporation of both gate and attention mechanisms in the regular graph convolution.

| Model | TWITTER | | LAP14 | | REST14 | | REST15 | | REST16 | |
|---|---|---|---|---|---|---|---|---|---|---|
| | Acc | F1 | Acc | F1 | Acc | F1 | Acc | F1 | Acc | F1 |
| CKG LSTM w/o a | 74.12 | 72.25 | 77.09 | 73.01 | 82.14 | 73.07 | 82.35 | 65.77 | 89.88 | 71.04 |
| CKG LSTM w/o ga | 73.69 | 71.01 | 73.81 | 69.51 | 80.83 | 70.03 | 81.10 | 60.45 | 88.74 | 67.36 |
| CKG LSTM w/o a+ga | 73.21 | 70.62 | 73.49 | 68.07 | 80.65 | 69.76 | 80.01 | 58.14 | 87.58 | 64.32 |
| CKG LSTM | 75.83 | 73.02 | 79.80 | 75.57 | 84.40 | 76.01 | 84.27 | 70.50 | 91.02 | 71.89 |
| CKG BERT w/o a | 75.77 | 73.88 | 81.37 | 76.12 | 87.13 | 79.78 | 82.51 | 71.73 | 91.58 | 71.88 |
| CKG BERT w/o ga | 76.61 | 74.76 | 80.04 | 75.64 | 87.96 | 81.97 | 88.32 | 75.64 | 91.72 | 75.62 |
| CKG BERT w/o a+ga | 76.27 | 74.40 | 80.33 | 75.33 | 87.28 | 79.41 | 86.87 | 70.31 | 90.08 | 74.71 |
| CKG BERT | **77.95** | **77.20** | **83.41** | **79.41** | **89.24** | **83.31** | **88.92** | **76.42** | **92.87** | **81.95** |

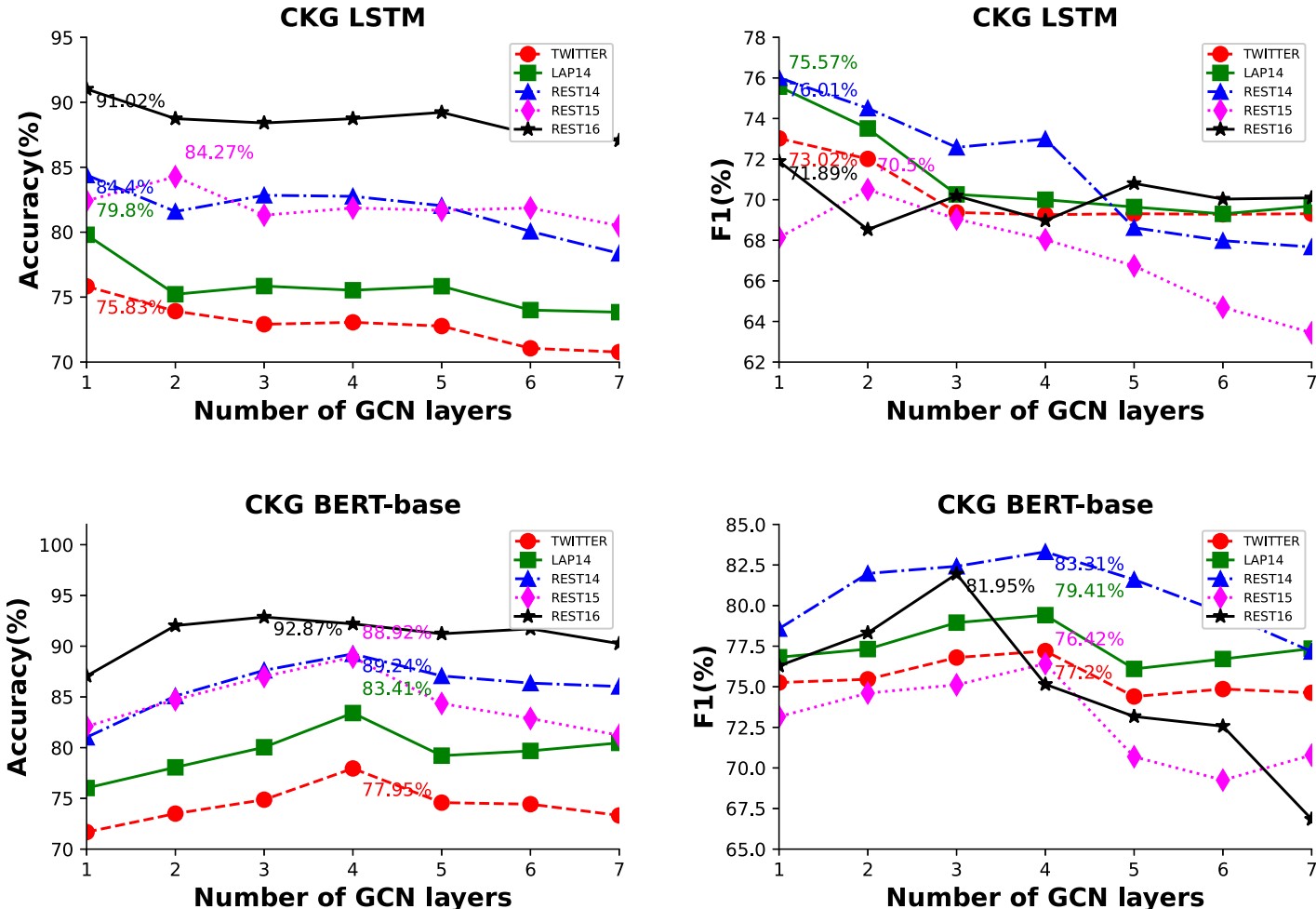

**Fig 9. The ACC and F1 values obtained by training and testing with different CKG layer settings after applying ChatGPT data augmentation on the five datasets (Twitter, LAP14, REST14, REST15, and REST16).**

incorporating the "BERT" encoding consistently outperform those using "LSTM". Additionally, the models without attention or gate mechanisms exhibit unsatisfactory performance across all datasets, indicating the significant improvement in model performance with the inclusion of attention mechanisms and the gate-augmented GCN ("GAGCN") for regular GCN. Furthermore, it is noteworthy that models with only attention mechanisms or only gate mechanisms perform worse than the models that incorporate both.

Based on the above analysis, we can conclude that our proposed enhancements effectively integrate semantic and syntactic features. The model, when combined with the sentiment knowledge base, demonstrates outstanding performance and merits attention in ABSA tasks.

## 4.6 Effect of the GCN layer number

The impact of the number of GCN layers on the focus of aspect features with varying degrees of local contextual information is often observed. We investigated the influence of the number of GCN layers on model performance, as shown in Fig 9. The evaluation was conducted using ACC and F1 scores, with the GCN layers ranging from 1 to 7. We tested the models on the five

datasets using either the BiLSTM with attention mechanism as the semantic module or the BERT-base encoding. The graph convolution modules used in this study were based on the proposed GAGCN network.

It is apparent that there exists a notable disparity in the outcomes when employing BERT-base as the semantic extraction module compared to the absence thereof. Under the same number of GCN layers, models using BERT-base as the encoding module outperform those using BiLSTM. When not using the BERT-base module, the model performs best with 1 to 2 GCN layers. One possible reason is that the dimensionality of the data encoded by BiLSTM is smaller, and GCN can learn the syntactic features with fewer layers. When BERT-base is used as the sentence encoding module, on the one hand, the model has a better understanding of semantics, and on the other hand, the input dimensionality of the syntax module is larger. Therefore, the model performs best with 3 to 4 GCN layers. One common observation for both encoding methods is that the model's performance tends to decline when the number of layers exceeds a certain threshold, indicating overfitting due to the complexity of the model.

## 4.7 Attention visualization

By visualizing the attention weights, we can intuitively observe how the CKG model focuses on different sentiment words to extract specific aspect-level sentiment features. As shown in Fig 10, darker colors indicate higher attention weights given by the model to those words. From the upper subgraph of Fig 10, we can observe that a general attention model exhibits some level of attention towards the aspect words in the sentence. However, it also assigns weights to other irrelevant words. Moreover, when there are multiple aspect words in a single sentence, the model fails to distinguish them significantly. On the other hand, the lower subgraph demonstrates that the CKG model, after adjustment, exhibits clear attention towards each aspect word and the corresponding words that influence the sentiment polarity. It assigns less attention to other words. Based on the aforementioned observations, it can be deduced that the model proposed in this study exhibits formidable prowess in addressing Aspect-Based Sentiment Analysis (ABSA) and presents commendable interpretability.

## 4.8 Case study

Through Table 7, we can observe the model's performance during actual predictions. Examples 1 to 3 are selected from the test set of the dataset, while examples 4 and 5 are comment-like statements generated by ChatGPT. By examining examples 1, 2, 4, and 5, we can see that our model accurately predicts the sentiment tendencies corresponding to different aspect words in sentences that contain multiple aspects. Additionally, our model demonstrates robustness by accurately predicting sentiment in the simulated comment-like statements that align with human conventions. Example 3 showcases our model's ability to accurately predict sentiment for aspect words associated with neutral emotions, further highlighting its excellent performance.

## 5. Conclusion

In this paper, we use ChatGPT to enhance text data and we modify the network architecture to improve the model's understanding of fine-grained emotions. Building upon the traditional "semantic & syntactic" analysis paradigm, we further improve it by introducing a cross-fusion mechanism that effectively integrates semantic and syntactic features. Additionally, we introduce a gated attention mechanism in the graph convolutional network to improve the performance of the syntactic feature extraction module and alleviate the potential issue of gradient explosion associated with multi-layer graph convolutions. We also incorporate an external

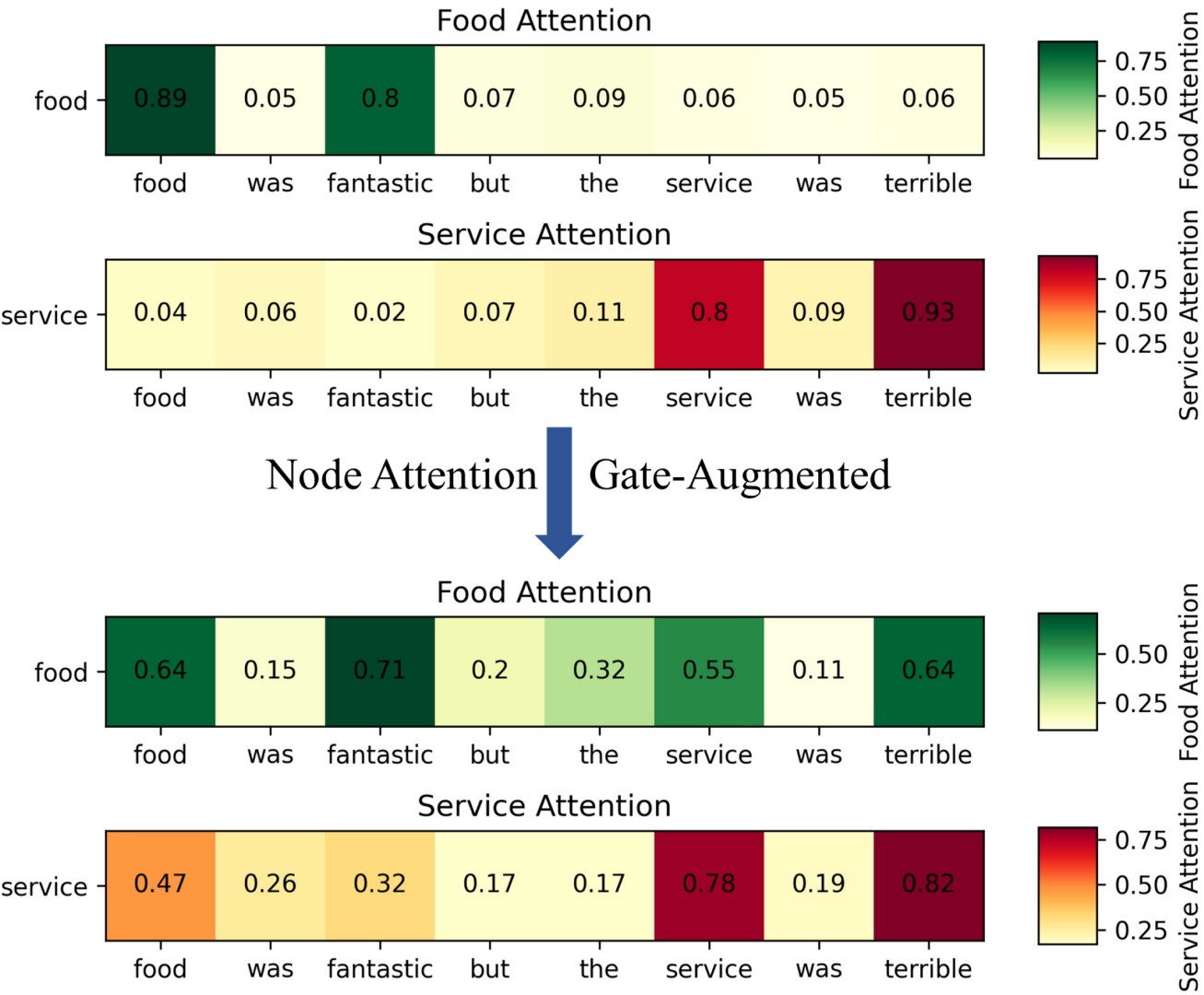

**Fig 10. The attention visualizations of typical samples.**

**Table 7. The scoring criteria for evaluating the applicability of text data generated by ChatGPT.**

| Sample | Sentence | Sentiment | Predict |
|---|---|---|---|
| 1 | love the drinks, esp lychee martini, and the food is also VERY good | pos, pos, pos | pos, pos, pos |
| 2 | i liked the atmosphere very much but the food was not worth the price | pos, neg | pos, neg |
| 3 | the decor is rustic, traditional japanese. | neu | neu |
| 4 | The hotel room was spacious and clean, but the Wi-Fi connection was slow and unreliable. | pos, neg | pos, neg |
| 5 | The movie had breathtaking visual effects, but the plot was confusing and hard to follow. | pos, neg | pos, neg |

knowledge base to enhance the characteristics of aspect words, ensuring that the model maintains its focus on aspect words. This enhancement significantly improves the accuracy and interpretability of the model's predictions. In summary, our work makes considerable advancements in ABSA for neurorobotics. In the future, we will continue to explore the potential of ChatGPT, focusing on expanding and enriching large-scale datasets for ABSA. Furthermore, we plan to refine and simplify the model architecture while delving into the domain of multilingual analysis.

## Author Contributions

**Conceptualization:** Yapeng Gao.

**Data curation:** Yapeng Gao.

**Investigation:** Yapeng Gao.

**Methodology:** Yapeng Gao.

**Supervision:** Yapeng Gao, Lin Zhang, Yangshuyi Xu.

**Validation:** Yapeng Gao.

**Visualization:** Yapeng Gao.

**Writing – original draft:** Yapeng Gao.

**Writing – review & editing:** Yapeng Gao.

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
