## [Decision Letter · Decision Letter 0]

3 Nov 2023

PONE-D-23-24706Improving Aspect-Based Sentiment Analysis with Affective Knowledge Enhancements and Graph Adaptive Attention MechanismPLOS ONE

Dear Dr. Zhang,

Thank you for submitting your manuscript to PLOS ONE. After careful consideration, we feel that it has merit but does not fully meet PLOS ONE’s publication criteria as it currently stands. Therefore, we invite you to submit a revised version of the manuscript that addresses the points raised during the review process.

We look forward to receiving your revised manuscript.

Kind regards,

Toqir Rana, Ph.D.

Academic Editor

PLOS ONE

5. We are unable to open your Supporting Information file [reference.bib]. Please kindly revise as necessary and re-upload.

Reviewers' comments:

Reviewer's Responses to Questions

**Comments to the Author**

1. Is the manuscript technically sound, and do the data support the conclusions?

Reviewer #1: Partly

Reviewer #2: Yes

2. Has the statistical analysis been performed appropriately and rigorously? 

Reviewer #1: Yes

Reviewer #2: I Don't Know

3. Have the authors made all data underlying the findings in their manuscript fully available?

Reviewer #1: Yes

Reviewer #2: Yes

4. Is the manuscript presented in an intelligible fashion and written in standard English?

Reviewer #1: Yes

Reviewer #2: No

5. Review Comments to the Author

Reviewer #1: The author proposed method for Improving Aspect-Based Sentiment Analysis with Affective Knowledge Enhancements and Graph Adaptive Attention Mechanism mainly used Graph Convolutional Neural Networks (GAN). The experimental analysis is not well explained, and the proposed method is worthy for investigation. The paper is lack of small issue which should be consider for improving the manuscript.

1. In Abstract section not proper clarification about Problem statement

2. Author mainly focused Dataset, features, and metrics in whole paper, so that it is missing Methods clarification properly, kindly re organized paper.

3. The author must focus on the small typos, uses of punctuation and English level throughout the manuscript.

4. I will suggest to author to use proper table for comparing techniques with some relevant parameters.

5. I will suggest author to use more exiting method for comparison of the proposed method to show the result.

6. Need to modify references also, most of the references are too old.

7. Author Should add Future scope of this Article.

Reviewer #2: 1. There are no results in the abstract. Abstract must contain the important results.

2. There are typos and other language issues in the text. For example, “food”. There must be opening and closing quotes, whereas it contains only closing quotes on both sides of the word food.

3. Line 173. It should be Figure 3, not just 3.

4. Line 199, “it contains n words.” This is wrong, it contains n + 1 words, since the subscript is starting from 0 not 1.

5. The formulas at line 232 have no citation. Their source must be cited in case the authors themselves have not come up with them.

6. Equation 9, there must be space between contains and dependency.

7. The conclusion of the study must be furnished with the concrete results.

8. A discussion section needs to be added in the study to compare and contrast your results with the published works.

9. What are the limitations of your study?

10. Future work needs to be added in this work.

6. PLOS authors have the option to publish the peer review history of their article (what does this mean?). If published, this will include your full peer review and any attached files.

Reviewer #1: **Yes: **ZULFIKAR ALI ANSARI

Reviewer #2: **Yes: **Dr Nadeem Iqbal

---

## [Author Response · Author response to Decision Letter 0]

7 Jan 2024

Dear Editor,

Thank you for your letter and the reviewers’ comments concerning our manuscript. We sincerely appreciate receiving your letter and the reviewers' comments regarding our manuscript titled "Improving Aspect-Based Sentiment Analysis with Affective Knowledge Enhancements and Graph Adaptive Attention Mechanism"(PONE-D-23-24706R1). This research paper is now entitled “CKG: Improving Aspect-Level Sentiment Analysis with Text Augmentation using ChatGPT and Knowledge-Enhanced Gated Attention Graph Convolutional Networks”. Your valuable comments have been instrumental in guiding us towards the revision and improvement of our paper. We have thoroughly examined each comment and made the necessary revisions accordingly. We have reconstructed the article and virtually rewritten the entire paper. By the way, we have deleted the “reference.bib” file.

We would like to express our gratitude once again for your patience and guidance throughout this process.

Best regards,

Yapeng Gao, Lin Zhang, Yangshuyi Xu

Reviewer's Responses to Questions

Dear Reviewer #1,

Thank you for reviewing our paper and providing valuable feedback. We appreciate your constructive comments and suggestions. We have almost completely rewritten the entire article. We have carefully considered each point raised and have made the following revisions to address your concerns:

1. We have rewritten the abstract section of the original manuscript, summarizing the issues present in the field of aspect-level sentiment analysis from previous research and our corresponding solutions in the revised Abstract section.

2. We have reorganized the paper to provide a more balanced explanation of the proposed method, giving appropriate attention to the methods used.

3. We have systematically reviewed and addressed the improper word usage and grammar errors throughout the article. It is worth mentioning that we have almost completely rewritten the entire paper.

4. In section 4.4 "Results," we have included Table 5, which compares the performance of our model with some other models in the field of aspect-level sentiment analysis. This table provides a comprehensive comparison of our model's performance. In section 4.5 "Ablation Study," we have presented Table 6, which visually presents the results of our model's ablation experiments. Additionally, we have provided explanations regarding the configuration of model-related layers and encoding choices.

5. To enhance the comparison of the proposed method, we have incorporated more exciting methods and presented the results accordingly. In the section 4.4 "Results" of the paper, we have compared our model with traditional and recent mainstream models, using commonly accepted evaluation metrics in the field of aspect-level sentiment analysis. We have analyzed the comparative results and provided insights based on these evaluation standards, which have achieved a consensus in the field. In the section 4.7 "Attention Visualization" and the section 4.8 "Case Study", we have further elaborated on the effectiveness of our model using a combination of images and textual explanations. These sections provide visual and descriptive evidence to support the efficacy of our model.

6. We have revised the references and included more recent sources to ensure the relevance and up-to-dateness of the cited literature.

7. In section 5 "Conclusion", we have supplemented the future research directions.

Once again, we thank you for your valuable feedback, which has significantly improved the quality of our paper. We hope that the revisions adequately address your concerns and improve the overall clarity and presentation of our work.

Dear Reviewer #2,

Thank you for your thorough review of our paper. We appreciate your valuable feedback and have carefully considered each of your points. We have almost completely rewritten the entire article. We have made the following revisions to address your concerns:

1. Based on your feedback, we have restructured the Abstract section, highlighting the existing shortcomings in the past research and our research focus. We have ensured a clear and concise correspondence between them.

2. We apologize for such a basic mistake. We have rectified this error and revised the article.

3. We have conducted a detailed review of the image labels throughout the entire document, ensuring that each image is numbered clearly and accurately.

4. You are absolutely right, and I apologize for the mistake. We have meticulously derived all the formulas in the document, ensuring their accuracy. The corrected formulas can be found on line 246.

5. We have provided the sources for all the formulas that were not originally proposed by us.

6. We have adjusted the expression of this formula (Eq3) and reviewed other text-containing formulas to ensure their accuracy.

7. We have furnished the conclusion with concrete results to provide a more comprehensive summary of our study.

8. Firstly, in the Results section (4.4), we have provided a demonstration of the effectiveness of our experimental results. Secondly, in sections 4.5 to 4.8, we have discussed and analyzed our ablation experiments and the improvement achieved by our model. We have also included examples of practical applications.

9. The usage limit of ChatGPT can indeed affect our research efficiency. As mentioned in the paper, the "expert review of text generation quality" required communication and incurred a certain time cost. Additionally, even with specified evaluation metrics, human subjectivity can still influence the final judgment. Therefore, we are actively considering an effective and quantifiable evaluation method. In the Conclusion section (5), we elaborate on this matter.

10. In the Conclusion section of the paper, we have provided an exposition on the future research directions.

We appreciate your feedback, which has significantly helped improve the quality and clarity of our paper. We hope that the revisions adequately address your concerns and enhance the overall contribution of our work. Thank you once again for your valuable input.

---

## [Decision Letter · Decision Letter 1]

25 Jan 2024

PONE-D-23-24706R1CKG: Improving Aspect-Level Sentiment Analysis with Text Augmentation using ChatGPT and Knowledge-Enhanced Gated Attention Graph Convolutional NetworksPLOS ONE

Dear Dr. Zhang,

Thank you for submitting your manuscript to PLOS ONE. After careful consideration, we feel that it has merit but does not fully meet PLOS ONE’s publication criteria as it currently stands. Therefore, we invite you to submit a revised version of the manuscript that addresses the points raised during the review process.

We look forward to receiving your revised manuscript.

Kind regards,

Toqir Rana, Ph.D.

Academic Editor

PLOS ONE

Journal Requirements:

Reviewers' comments:

Reviewer's Responses to Questions

**Comments to the Author**

1. If the authors have adequately addressed your comments raised in a previous round of review and you feel that this manuscript is now acceptable for publication, you may indicate that here to bypass the “Comments to the Author” section, enter your conflict of interest statement in the “Confidential to Editor” section, and submit your "Accept" recommendation.

Reviewer #1: All comments have been addressed

Reviewer #2: (No Response)

2. Is the manuscript technically sound, and do the data support the conclusions?

Reviewer #1: Yes

Reviewer #2: Yes

3. Has the statistical analysis been performed appropriately and rigorously? 

Reviewer #1: Yes

Reviewer #2: Yes

4. Have the authors made all data underlying the findings in their manuscript fully available?

Reviewer #1: Yes

Reviewer #2: Yes

5. Is the manuscript presented in an intelligible fashion and written in standard English?

Reviewer #1: Yes

Reviewer #2: Yes

6. Review Comments to the Author

Reviewer #1: All changes have been made. There is no need for further changes from my end. Now manuscript is OK to publish.

Reviewer #2: 1. Page 10. ABSA. This abbreviation has been used but I could not find its full form. It must be given in full form first time. Then its abbreviated version will be used. In the same way, other abbreviations must be treated.

2. There are language and other grammar issues in the manuscript which must be addressed.

3. Page 15. Equations must be cited if authors have not developed them.

4. Algorithm 1. “nd”. I think it is “and”

5. Line 285. “algorithm” must be changed with “Algorithm”. Since a specific algorithm is being referred to.

6. Line 306. Table1. It needs space.

7. Page 17. Equation 3. “D” should be made italic. All the variables being used must be italicized.

7. PLOS authors have the option to publish the peer review history of their article (what does this mean?). If published, this will include your full peer review and any attached files.

Reviewer #1: **Yes: **ZULFIKAR ALI ANSARI

Reviewer #2: **Yes: **Dr Nadeem Iqbal

---

## [Author Response · Author response to Decision Letter 1]

28 Feb 2024

Dear Editor,

Thank you for your letter and the reviewers’ comments concerning our manuscript. We sincerely appreciate receiving your letter and the reviewers' comments regarding our manuscript titled " CKG: Improving ABSA with Text Augmentation using ChatGPT and Knowledge-Enhanced Gated Attention Graph Convolutional Networks" (PONE-D-23-24706R1). Your valuable comments have been instrumental in guiding us towards the revision and improvement of our paper. We have thoroughly examined each comment and made the necessary revisions accordingly. 

We would like to express our gratitude once again for your patience and guidance throughout this process.

Best regards,

Yapeng Gao, Lin Zhang, Yangshuyi Xu

Reviewer's Responses to Questions

Dear Reviewer #1,

Thank you for your recognition of our research. We are confident that with your meticulous academic approach, you will undoubtedly be able to generate further achievements.

Dear Reviewer #2,

Thank you once again for your valuable feedback. Your meticulous attention to detail has deeply touched us. Based on your suggestions, we have made the following revisions:

1. We have included an explanation of the abbreviation ABSA in the abstract, and replaced the existing content throughout the entire manuscript while also verifying other abbreviations.

2. We have made modifications to the grammar and expressions in the manuscript, and have indicated these changes in the "Revised Manuscript with Track Changes."

3. We have indicated the source of the equations and acknowledge your reminder.

4. Thank you for your attention to detail. We have made the necessary correction to the erroneous expression, replacing "nd" with "and".

5. We have corrected the initial term "algorithm" to "Algorithm" and reviewed other similar usage for any potential errors.

6. At the corresponding locations, we have populated the empty spaces.

7. Based on your feedback, we have made adjustments to all the formulas by converting them into italics format.

Thank you for acknowledging our research work, which motivates us to pursue further investigations. With your guidance, we believe this paper can yield more remarkable outcomes and reach a wider scholarly audience. Wishing you continued success in your endeavors.

---

## [Decision Letter · Decision Letter 2]

18 Mar 2024

CKG: Improving Aspect-Level Sentiment Analysis with Text Augmentation using ChatGPT and Knowledge-Enhanced Gated Attention Graph Convolutional Networks

PONE-D-23-24706R2

Dear Dr. Zhang,

We’re pleased to inform you that your manuscript has been judged scientifically suitable for publication and will be formally accepted for publication once it meets all outstanding technical requirements.

Kind regards,

Toqir Rana, Ph.D.

Academic Editor

PLOS ONE

Additional Editor Comments (optional):

Authors are requested to incorporate changes highlighted by the reviewer#2 before the final submission.  

Reviewers' comments:

Reviewer's Responses to Questions

**Comments to the Author**

1. If the authors have adequately addressed your comments raised in a previous round of review and you feel that this manuscript is now acceptable for publication, you may indicate that here to bypass the “Comments to the Author” section, enter your conflict of interest statement in the “Confidential to Editor” section, and submit your "Accept" recommendation.

Reviewer #2: All comments have been addressed

2. Is the manuscript technically sound, and do the data support the conclusions?

Reviewer #2: Yes

3. Has the statistical analysis been performed appropriately and rigorously? 

Reviewer #2: Yes

4. Have the authors made all data underlying the findings in their manuscript fully available?

Reviewer #2: Yes

5. Is the manuscript presented in an intelligible fashion and written in standard English?

Reviewer #2: Yes

6. Review Comments to the Author

Reviewer #2: Following are my reviews:

1. Algorithm 1 inputs are D and Dn. but this algorithm is using many other data structures like POS. How are they accessible to this algorithm?

2. It would be nice if abstract is furnished with the important results, this study has found.

7. PLOS authors have the option to publish the peer review history of their article (what does this mean?). If published, this will include your full peer review and any attached files.

Reviewer #2: **Yes: **Dr Nadeem Iqbal
